# In-Silico Analysis of Monoclonal Antibodies against SARS-CoV-2 Omicron

**DOI:** 10.3390/v14020390

**Published:** 2022-02-14

**Authors:** Ye-Fan Hu, Jing-Chu Hu, Hin Chu, Thomas Yau, Bao-Zhong Zhang, Jian-Dong Huang

**Affiliations:** 1School of Biomedical Sciences, Li Ka Shing Faculty of Medicine, University of Hong Kong, Hong Kong, China; yefanhu@connect.hku.hk; 2Department of Medicine, Li Ka Shing Faculty of Medicine, University of Hong Kong, Hong Kong, China; tyaucc@hku.hk; 3CAS Key Laboratory of Quantitative Engineering Biology, Shenzhen Institute of Synthetic Biology, Shenzhen Institutes of Advanced Technology (SIAT), Chinese Academy of Sciences, Shenzhen 518055, China; jc.hu@siat.ac.cn; 4Department of Microbiology, Li Ka Shing Faculty of Medicine, University of Hong Kong, Hong Kong, China; hinchu@hku.hk; 5Guangdong-Hong Kong Joint Laboratory for RNA Medicine, Sun Yat-Sen University, Guangzhou 510120, China

**Keywords:** SARS-CoV-2, omicron, antibody, vaccine

## Abstract

Omicron was designated by the WHO as a VOC on 26 November 2021, only 4 days after its sequence was first submitted. However, the impact of Omicron on current antibodies and vaccines remains unknown and evaluations are still a few weeks away. We analysed the mutations in the Omicron variant against epitopes. In our database, 132 epitopes of the 120 antibodies are classified into five groups, namely NTD, RBD-1, RBD-2, RBD-3, and RBD-4. The Omicron mutations impact all epitopes in NTD, RBD-1, RBD-2, and RBD-3, with no antibody epitopes spared by these mutations. Only four out of 120 antibodies may confer full resistance to mutations in the Omicron spike, since all antibodies in these three groups contain one or more epitopes that are affected by these mutations. Of all antibodies under EUA, the neutralisation potential of Etesevimab, Bamlanivimab, Casirivimab, Imdevima, Cilgavimab, Tixagevimab, Sotrovimab, and Regdanvimab might be dampened to varying degrees. Our analysis suggests the impact of Omicron on current therapeutic antibodies by the Omicron spike mutations may also apply to current COVID-19 vaccines.

## 1. Introduction

During the current Coronavirus Disease 2019 (COVID-19) pandemic caused by severe acute respiratory syndrome coronavirus 2 (SARS-CoV-2), the World Health Organization (WHO) has been tracking SARS-CoV-2 variants in terms of variants of concern (VOCs), variants of interest (VOIs), or variants under monitoring (VUMs) [1]. Previous VOCs including Alpha (Pango lineage B.1.1.7), Beta (B.1.351), Gamma (P.1), and Delta (B.1.617.2 and AY.*) were usually designated 3 to 6 months after they were first reported. The most recent VOC, Omicron (B.1.1.529 or BA.*), was named on 26 November 2021, only 4 days after its sequence was first submitted [2]. The urgent action was prompted by the identification of an unusually large number of mutations in the Omicron spike, which includes 10 mutations in the N-terminal domain (NTD) and 15 mutations in the receptor binding domain (RBD) [3]. While such mutation numbers delayed the validation of impact on therapeutic antibodies and vaccines.

Past studies have isolated hundreds of antibodies against the SARS-CoV-2 spike protein for analysing epitopes or developing therapeutic drugs. This allows the identification of the precise structures of antigen-antibody complexes. These studies offered reliable data to estimate the impact of emerging viral variants instantly on vaccine evasion without experiments. To estimate the impact, we collected 132 confirmed epitopes of 120 monocloncal antibodies (Appendix A) targeting five major antigenic groups, namely NTD [4], RBD-1, RBD-2, RBD-3, and RBD-4 [5]. Since the spike protein in vaccines approved shares an identical structural basis for generating the above-mentioned antibodies, it is urgent to analyse the impact of all mutations in the Omicron spike on both vaccines and therapeutic antibodies.

Here, we evaluate the impact of Omicron spike mutations on vaccines and antibodies using our SARS-CoV-2 spike antibody database. Our analysis shows that the Omicron mutations impact all epitopes in NTD, RBD-1, RBD-2, and RBD-3, with no antibody-binding sites spared by these mutations. Only four antibodies in RBD-4 may confer full resistance to mutations in the Omicron spike. Of all antibodies under EUA, neutralisation potential of Etesevimab, Bamlanivimab, Casirivimab, Imdevima, Cilgavimab, Tixagevimab, Sotrovimab, and Regdanvimab might be dampened to varying degrees. Our analysis suggests the impact of Omicron on current therapeutic antibodies by the Omicron spike mutations may also apply to current COVID-19 vaccines.

## 2. Methods

We collected 132 confirmed conformational epitopes with protein structures released in the Protein Data Bank (PDB) or annotated epitope footprints in the literature. For antibodies with protein structures, the epitope residues were calculated following the IEDB method [6]; otherwise, partially epitope positions were collected from reference results. Details of all antibodies with epitope footprints, PDB access numbers, and reference DOI number are shown in Appendix A. Some antibodies show different epitopes in different studies, and we documented them as different epitopes. We plotted the spike protein epitopes using Microsoft Powerpoint.

In our database, 132 monoclonal antibody epitopes can be classified into five antigenic groups: NTD [4], RBD-1, RBD-2, RBD-3, and RBD-4 [5]. Among these epitopes, 19 epitopes recognised by 19 antibodies are in the NTD antigenic group, while in RBD, 114 epitopes bound by 102 antibodies targeting the RBD of spike protein. There are 42, 35, 22, and 14 documented epitopes for the antigenic group RBD-1, RBD-2, RBD-3, and RBD-4, respectively. Here, NTD contains epitopes covering residues 14–20, 27–30, 32, 61, 64, 66, 68, 69, 71, 76, 77, 97, 98, 124, 140, 142–158, 180–183, 185–187, 211–218, 243–257, 259, 260, and 262 of the spike protein; RBD-1 contains epitopes covering residues 403–408, 414–417, 420, 421, 432, 439- 441, 443–450, 452, 453, 455–460, 470–479, 481–496, 498–505, and 508; RBD-2 contains epitopes covering residues 372, 403, 405, 406, 408, 409, 414–417, 420, 421, 440, 444–446, 449, 450, 453, 455–460, 470, 471, 473–478, 481–487, 489, 490, 492–496, 498, and 500–505; RBD-3 contains epitopes covering residues 333–335, 337, 339–347, 349, 351, 354–361, 367, 368, 371–376, 378, 408, 409, 414, 417, 436–439, 440–452, 455, 456, 470, 472, 473, 475, 478–494, 498–506, 508, and 509; and RBD-4 contains epitopes covering residues 353–357, 359, 360, 366, 369–372, 374–386, 388–390, 392, 394, 396, 404, 405, 408, 409, 412–416, 426–430, 437, 462–466, 468, 500–506, 508, 514–521, and 523.

We used the earliest reported Omicron Spike sequences as the reference (GISAID accession number EPI_ISL_6590608 and EPI_ISL_6754457) [2]. In our analysis, an antibody completely resists the mutations in the Omicron spike if no mutations are identified at the epitopes. Otherwise, the antibody may be impacted.

## 3. Results

Mutations in the Omicron spike are potentially associated with a decrease on the effectiveness of current vaccines and antibody-based therapeutics. In our SARS-CoV-2 monoclonal antibody database with 132 structure-confirmed conformational epitope information of 120 antibodies (Appendix A), only four antibodies may confer full resistance to the mutations in the Omicron spike, as no mutations were identified at the epitopes bound by these antibodies. The monoclonal antibody epitopes can be classified into five antigenic groups: NTD [4], RBD-1, RBD-2, RBD-3, and RBD-4 [5] (Figure 1A–E).

We analysed the mutations in the Omicron spike against these previously identified epitopes. Our results suggest that the mutations in the Omicron spike dramatically impact NTD, RBD-2, and RBD-3, since all antibodies in these three groups contain one or more epitopes that are affected by these mutations (Figure 1A–D). There are 2.00 (standard error, s.d. 0.92), 3.29 (s.d. 1.79), and 5.86 (s.d. 1.22) mutations in these groups on average, while the average mutation number is 1.14 (s.d. 1.19) in RBD-4 for each epitope. Importantly, 4 antibodies targeting RBD-4 show full resistance to the mutations in the Omicron spike (Figure 1E). Three of them, including S304 [8], COVOX-45 [9], and S2H97 [10], contain epitopes that are not affected by mutations in the Omicron spike. Most of these antibodies have been demonstrated to have a high potency of cross-reactivity against multiple SARS-CoV-2 VOCs, SARS-CoV-1, or even pan-sarbecovirus. However, the other one, C126, is not a neutralising antibody, though it can bind to several variants of concern spike proteins [11]. Despite this, it is interesting that all antibodies with complete resistance to the Omicron mutations target the RBD-4 antigenic group.

Of all antibodies under emergency use authorization (EUA), our database suggests that the neutralisation potential of Etesevimab (LY-CoV016) [12], Bamlanivimab (LY-CoV555) [13], Casirivimab (REGN10933), Imdevimab (REGN10987) [14], Cilgavimab (AZD1061), Tixagevimab (AZD8895) [15], Sotrovimab (Vir-7831 or S309) [8,16], and Regdanvimab (CT-P59) [17] might be dampened to varying degrees due to the extensive mutations in the Omicron spike (Figure 1F). A past study reported a distinct impact of a single mutation in the RBD, and some antibodies were influenced by a single mutation dramatically [18]. For example, only K417N dampens the neutralising activity of Etesevimab, and E484A weakens Bamlanivimab significantly. The neutralisation induced by Imdevimab is destroyed by G446S solely [18]. Other antibodies are influenced by multiple mutations, while the combinatorial effect of these mutation can only be estimated based on single mutation’s effect. Regdanvimab might be dampened by K417N, E484A, Q493R and Y505H in Omicron together. Neutralisation induced by Casirivimab might be affected by K417N, E484A, and Q493R in Omicron collectively, though each mutation changes antigenicity moderately. S477N, T478K, and E484A might slightly influence Tixagevimab. Fortunately, two antibodies may only be influenced by the mutations in the Omicron spike slightly. Sotrovimab might be weaken by G339D and N440K only. Cilgavimab might be affected by N440K and G446S moderately. It is urgent to update current therapeutic antibody cocktails to fight the Omicron variant.

## 4. Discussion

In this study, we analysed mutations in the Omicron variant against 132 epitopes of 120 known antibodies. Our analysis suggested that four out of five antigenic groups are impacted by Omicron mutations. Only four antibodies in the RBD-4 antigenic group may remain fully resistant to mutations in the Omicron spike. Since the spike protein in all the emergency use listing (EUL) vaccines approved by WHO [19] shares a similar/identical structural basis for inducing the above-named antibodies, our analysis suggests the impact of Omicron on current therapeutic antibodies by the Omicron spike mutations may also apply to current COVID-19 vaccines.

Our analysis provides a novel insight for viral escape estimation when facing a newly emerged variant of SARS-CoV-2. Past studies utilised neutralisation experiments to validate variant escape. Experimental validations used to require a few weeks to obtain precise results. Such a delay might be detrimental for the public during the pandemic. Our analysis can offer rapid and accurate predictions to eliminate public panic. However, past variants only affected one or two vital mutations in the spike, which limits deep understanding of the combinatorial effect of mutations. In our analysis, it is difficult to provide a comprehensive analysis of the combinatorial effect. For more precise estimation, the combinatorial effect of multiple mutations should be modelled in the future when relevant data are available.

Our analysis estimated the antigenic shift of the Omicron variant. In our database, there are five major antigenic groups in the spike protein of SARS-CoV-2. Since the mutations in the Omicron spike impacted four major antigenic groups completely, this indicated that the antigenicity of the Omicron spike dramatically changed when compared to the original strain of SARS-CoV-2. Past VOCs with antigenic drift influenced two antigenic groups at most [5]. The Alpha and Delta variant affected RBD-1 and RBD-3 respectively, while Beta and Gamma impacted on both RBD-1 and RBD-2 [5]. Here, the Omicron variant changed the antigenicity of NTD, RBD-1, RBD-2, and RBD-3 completely at the same time (Figure 1). Our result suggested that the antigenic shift generated by Omicron mutations created a spike protein with entirely new antigenic features. Despite our instant analysis, further detailed experiments should be performed to validate the antigenic drift or shift of Omicron as well as other effects of those mutations.

Our findings may provide rapid guidelines for the development of next-generation vaccines and therapeutic antibodies. The Omicron spike with multiple mutations shows high similarity to those in the artificially constructed neutralisation-resistant polymutant [20], which remarkably weakened protection from vaccines and past infection, and even vaccination post infection. Under the threat of Omicron or other future emerging variants, the development of next-generation vaccines or antibody-based therapies against SARS-CoV-2 should take into consideration effective humoral responses with broad neutralising activities against SARS-CoV-2 variants based on up-to-date viral evolution tracking. Our analysis showed all antibodies with complete resistance to the mutations in the Omicron spike target RBD-4 but no antibodies under EUA targets RBD-4 (Figure 1E). In the future, with cocktails of therapeutic antibodies it is better to use multiple antibodies against every major antigenic group to minimise the risk of the viral escape due to multiple mutations.

Importantly, it is critical to mine other therapeutic or protective antibodies targeting other antigens other than the SARS-CoV-2 spike protein. Even for the spike protein, most researchers focus on the RBD domain only. In our database (Appendix A), more than half of the antibodies (69 out of 120) merely target the RBD-1 or RBD-2 antigenic group. The over-focusing of antibodies targeting the spike protein or the RBD in the past may lead to the viral escape of Omicron in the present. It is possible that the evolution pressure on the spike protein results in the occurrence of the Omicron variant within one year after large-scale vaccination in multiple countries. It is better to pay more attention to other antigens, such as nucleocapsid or antigens from other open-reading-frames, in the future. Multiple antibodies targeting multiple antigens can minimize the potential antigenic shift or drift in emerging SARS-CoV-2 variants.

In addition, protective immunity from T cells should not be ignored, as evidence from healthcare workers [21] and T cell vaccines [22] suggest that T cell epitopes are also promising targets for inducing an immune response against SARS-CoV-2. T cell epitopes have been ignored in the development of vaccines against infectious diseases for a long time. In recent years, T cell epitopes has been utilised in cancer vaccines and showed promising efficacy in an early-stage clinical trial [23]. Since T cell epitopes are recognised without native conformation in the original antigens, T cell epitope-based vaccines can offer additional layers of protection against viral escape [21]. In the future, it is better to develop vaccines provoking both B cell and T cell immunity to protect against antigenic shift or drift of emerging variants.

In conclusion, our analysis rapidly estimated the impact of Omicron mutations on 120 antibodies. The Omicron mutations affected four out of five major antigenic group in the spike. Our findings further showed that four antibodies may remain effective against the SARS-CoV-2 Omicron variant. Eventually, our study may guide the future monitoring of variant antigenic evolution and the development of next-generation vaccines and therapeutic antibodies.

## Figures and Tables

**Figure 1 viruses-14-00390-f001:**
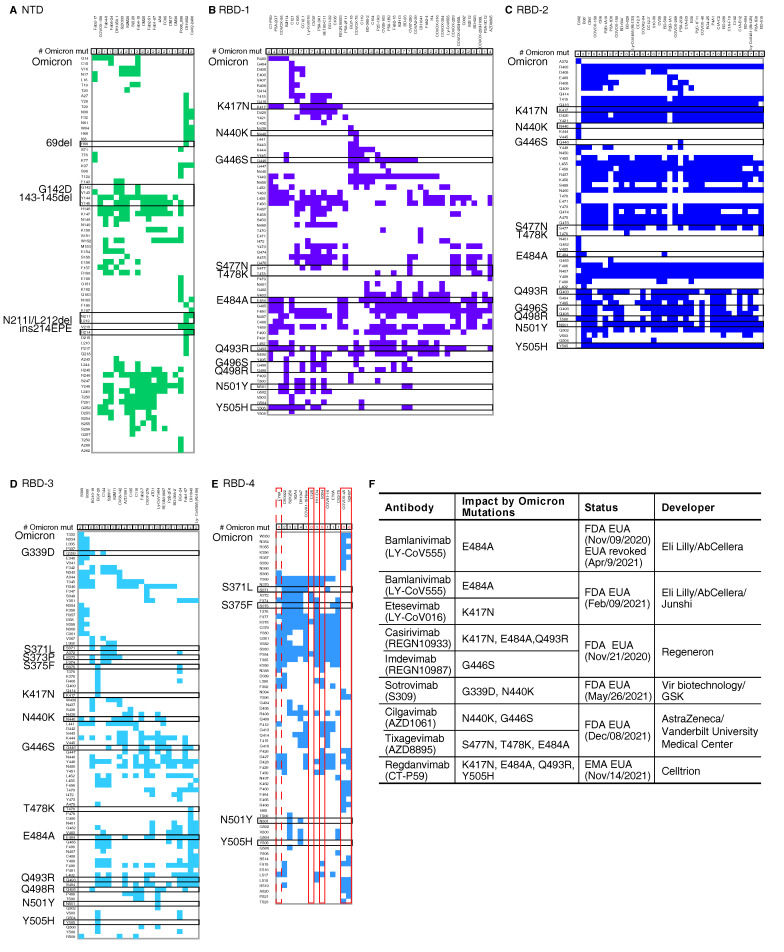
Mutations of SARS-CoV-2 Omicron and the epitopes of documented monoclonal antibodies. The epitopes on the five main antigenic sites (**A**) NTD, (**B**) RBD-1, (**C**) RBD-2, (**D**) RBD-3, and (**E**) RBD-4 are shown. The number of Omicron mutations (# Omicron mut) shows the number of mutations in each epitope. The black boxes indicate the Omicron spike mutations. The red boxes mark the antibodies that may retain effectiveness against Omicron. The dashed red box marks one type of EY6A [7] epitope that may have resistance. The classification of epitopes is based on Greaney AJ, et al. [5]. The epitopes and paired antibodies are listed in Appendix A. (**F**) Detailed information of therapeutic antibodies under emergency use authorization (EUA) analysed in this study.

## Data Availability

Publicly available datasets were analyzed in this study. This data can be found in Appendix A.

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
