# Peer review of "In-Silico Analysis of Monoclonal Antibodies against SARS-CoV-2 Omicron"

_viruses, 2022, doi:10.3390/v14020390_

Round 1
Reviewer 1 Report
An excellent paper in all respects. The title is a bit vague and I would suggest using monoclonal or therapeutic antibodies instead of just "antibodies".
Author Response
An excellent paper in all respects. The title is a bit vague and I would suggest using monoclonal or therapeutic antibodies instead of just "antibodies".
Responses: The title has been changed to ‘In-silico analysis of monoclonal antibodies against SARS-CoV-2 Omicron’.
Reviewer 2 Report
The authors analyzed the SARS-CoV-2 antibodies against Omicron which was useful for the current circulating variant. The manuscript can be improved further, I have several suggestions/queries:
- the title should be changed since the authors did not perform experiments, my suggestion: ‘In-silico analysis of antibodies against SARS-CoV-2 Omicron’
- line 36: for the first submitted sequence, the authors can refer to the study of Tsang et al. Unusual high number of spike protein mutations for the SARS-CoV-2 strains detected in Hong Kong. J Clin Virol. 2022 Jan 20;148:105081. doi: 10.1016/j.jcv.2022.105081. Epub ahead of print. PMID: 35091227.
- the introduction section should be concise and prevent duplication, you mentioned 121 antibodies three times!
(1) lines 1 to 40, up to ‘………. antibodies and vaccines’ were fine.
(2) re-organize the subsequent parts to describe the objectives only, the details can be put into methods section
- the method section has to be re-organized, it should consist of:
(1) number of antibodies and epitopes assessed (you did it)
(2) classification of epitopes groups (you mentioned in line 97, should put it here), move the paragraph (lines 72-91) to the method section
(3) elaborate how you come up for the epitope in Table S1
e.g. for antibody Fab4-18, according to the reference of Table S3, the number of residues mentioned (14, 15, 16, 17, 18, 19, 140, 142……..) were higher than your Table S1 (14, 15, 16, 17, 19, 142……..)
The number of residues mentioned in the reference were higher than yours in Table S1. You have to explain the rationale to list of the residues in your Table S1
(4) the assessment criteria to assess the effectiveness of antibodies (you mentioned in line 76, based on any mutations at the epitopes)
(5) cite the SARS-CoV-2 Omicron sequence you performed the assessment (the first sequence only?)
- the results section can be elaborated
(1) Figure 1 was good
(2) a table can be inserted to group the number mutations found in the epitopes (e.g. 0, 1-2, 3-4…….), the authors have to assess how these groupings can be arranged.
(3) lines 109-127, this paragraph was fine, use a separate table to focus the mutations of those EUA antibodies, then put the discussion of findings into discussion section.
- the discussion section was fine, however, the second paragraph have to elaborate more to alert the readers about your analysis, it was based on assessment of mutations found in epitope, the effect of those mutations were not known and further experiments should be performed
- minor issues about Table S1
(1) the order of the antibodies can be sorted according to the order shown in Figure 1, then readers can follow the details easily
(2) Table S1 can be divided into five sections according to Figure 1
(3) the reference used should be align with the same format shown in the main text, e.g. 1, 2, 3………etc. Then the references can be put under the table, this supplementary file can be treaded as a standalone file.
(4) check the latest citation used (the authors can set the date for latest access), for example Fab4-18 and Fab2-17 were come from the same reference. The citation of Fab4-18 was come from a pre-print article, the citation of Fab2-17 was from a pre-reviewed article. Indeed, these two references were the same.
Round 2
Reviewer 2 Report
The authors addressed all of my queries.